# MEMORY-PRUNING ALGORITHM FOR BAYESIAN OPTIMIZATION WITH STRICT COMPUTATIONAL COST GUARANTEES

## ABSTRACT

Bayesian Optimization (BO) is a powerful tool for optimizing noisy and expensive-to-evaluate black-box functions, widely used in fields such as machine learning and various branches of engineering. However, BO faces significant challenges when applied to large datasets or when it requires numerous optimization iterations. The computational and memory demands of updating Gaussian Process (GP) models can result in unmanageable computation times. To address these limitations, we propose a new Bayesian Optimization algorithm with memory pruning (MP-BO), which restricts the maximum training data size by acquiring new queries while concurrently removing data points from the training set. This approach guarantees a maximum algorithmic complexity of $\mathcal{O}(m^3)$, where $m \ll n$ is a fixed value and $n$ represent the size of the full training set. The pruning strategy ensures reduced and constant memory usage and computation time, without significantly degrading performance. We evaluate MP-BO on synthetic benchmarks and a real neurostimulation dataset, demonstrating its robustness and efficiency in scenarios where traditional BO would fail under strict computational constraints. Our results suggest that MP-BO is a promising solution for applications that require efficient optimization with limited computing resources.

## 1 INTRODUCTION

Bayesian optimization (BO) is used in a variety of applications to optimize costly to evaluate and noisy black-box functions. It has been widely applied in numerous fields, such as machine learning (Snoek et al., 2012) and several branches of engineering (Lam et al., 2018) due to its effectiveness in dealing with complex optimization problems with a small amount of data.

One of the most significant challenges of BO is its application to large volumes of training data. BO relies on constructing a model, called *surrogate*, to approximate an objective function and using an acquisition strategy to guide the search over a parameter space. The most commonly used surrogate models are Gaussian Processes (GPs) (MacKay, 1998). However, updating a GP with newly collected data points is computationally and memory-intensive. The algorithmic complexity of BO with GPs grows cubically with $n$, the number of past queries or collected data points. As a result, the usability of this method suffers as the amount of training data increases, leading to excessive computational times that are incompatible with time-sensitive problems. Furthermore, the memory requirements for data storage and processing grow quadratically with $n$, imposing significant constraints on the available memory resources (Kunjir, 2019).

These limitations are particularly challenging in closed-loop settings, where optimization must run on devices with constrained resources, such as embedded systems or small autonomous platforms. Many engineering fields rely heavily on autonomous decision-making to manage system dynamics and real-time operations. In many cases, strict requirements in decision-making time are imposed, which are incompatible with VANILLA BO increase in execution time, and a fixed limit is requred. These fields include autonomous robotics, where BO has compelling applications for learning and adaptation (Cully et al., 2015). Learning actuation patterns in real time, as the robot moves, requires performing optimization tasks within strict execution time limits and under constrained computational resources. In other domains, such as real-time financial trading systems, large computing

resources may be available, but rapid decision-making is essential, thus strict execution times must be enforced for algorithms to continuously trade on markets. Continuous increases in computing time for each action would make long-term continual optimization unfeasible.

Another field facing similar challenges is the development of intelligent medical devices. In this context, autonomous optimization and adaptation are desirable not only for robustness across environments of user's daily living, but also for ensuring patient data security. Implanted medical devices often must rely on low computational resources. One compelling application of BO in the realm of intelligent or *adaptive* (Beudel & Brown, 2016) medical devices is neurostimulation programming. In this context, a pacemaker-like device delivers stimuli to the brain or nervous system to evoke a desired physiological response, such as pain relief or improvement of motor control. The key challenge in neurostimulation is to efficiently identify the stimulation patterns and parameters, such as position, frequency, and intensity, that optimally evoke the targeted response. For example, BO can be used with deep brain stimulation to help treat Parkinson's disease (Sarikhani et al., 2022), to tune vagus nerve stimulation (Wernisch et al., 2024; Mao et al., 2024) and brain or spinal stimulation to recover walking after spinal cord injury (Wenger et al., 2014). BO is particularly advantageous in this context, often producing superior results compared to other search methods, even when exploring only a small subset of possible parameter combinations (Bonizzato et al., 2023; Laferrière et al., 2020). Minimizing computation time and memory usage is essential, as it directly influences the feasibility of system miniaturization. Compact, portable systems capable of being used outside the laboratory are critical for advancing clinical applications, and this development necessitates algorithms that are both highly performant and resource efficient.

To address these challenges, we propose a BO algorithm with a **M**emory-**P**runing method (MP-BO). Our approach iteratively deletes training data points as new queries are acquired, keeping the algorithmic complexity constant at $\mathcal{O}(m^3)$ for some chosen $m \ll n$. This pruning strategy not only alleviates memory constraints, but also drastically reduces the optimization time as $n$ increases. With MP-BO, we do not claim to outperform the classic BO algorithm, although, as we later demonstrate, there are cases where this is possible. Rather, our focus is on enforcing strict limits on computational time and memory usage while minimizing performance loss relative to full-capacity BO. Thus, MP-BO works by randomized eviction of training points, an effective choice that is agnostic to the problem structure and outperforms simple deterministic strategies.

Formally, we make the following contributions:

- We develop MP-BO, an algorithm that provides strict guarantees on memory usage and computational time by iteratively removing data from the training set at any time a new data point is acquired.
- We benchmarked our algorithm across various optimization problems, demonstrating its potential and assessing its robustness to noise level and increasing input size.
- We applied MP-BO to a real-world neurostimulation dataset, showcasing its effectiveness in a practical, real-world application.

## 2 BACKGROUND AND PROBLEM STATEMENT

In this section, we formally define the BO framework.

### 2.1 PROBLEM SETUP

We consider the problem of optimizing an unknown, or black-box function $f$ as follows:

$$\mathbf{x}^* = \underset{\mathbf{x} \in \mathcal{X}}{\operatorname{argmax}} f(\mathbf{x}) \tag{1}$$

such that $f : \mathcal{X} \to \mathbb{R}$ and $\mathcal{X} \subset \mathbb{R}^d$. Here, the *objective function* $f$ is expensive to evaluate.

We define the training dataset with $\mathcal{D}_{1:n} := (\mathbf{X}, \mathbf{y})$, where $\mathbf{X} = (\mathbf{x}_1, ..., \mathbf{x}_n)^T$ is the dataset of the points sampled in $\mathcal{X}$, and $\mathbf{y} = (y_1, ..., y_n)^T$ their corresponding observation. We deal with noisy observations, which means that we cannot directly access the objective function: $y_i = f(\mathbf{x}_i) + \epsilon_i \quad \forall i \in \{1, ..., n\}$, with $\epsilon_i \sim_{\text{iid}} \mathcal{N}(0, \sigma_{noi}^2)$. We only consider homoskedastic noise, where $\epsilon_i$ and $\mathbf{x}_i$ are independent, even if heteroskedastic noise can also be treated (Guzman et al., 2020).

## 2.2 BAYESIAN OPTIMIZATION

BO uses a *surrogate model* as a probability distribution for the objective function. Since the analytical form of the objective function is unknown, it is treated as a random function and assigned a belief *prior*. As the objective function is evaluated, BO calculates the new distribution *posterior*, using the *likelihood* of the observations and updating the prior using Bayes' theorem:

$$\underbrace{P(f|\mathcal{D}_{1:n})}_{\text{Posterior}} \propto \overbrace{P(\mathbf{y}|\mathbf{X}, f)}^{\text{Likelihood}} \overbrace{P(f|\mathbf{X})}^{\text{Prior}}. \tag{2}$$

The optimization is based on a sampling strategy which guides the algorithm in collecting the next point at each iteration. The sampling strategy is determined by maximizing an *acquisition function*, which provides a measure of utility for each possible next point to be sampled. In the beginning, if we have no prior knowledge about the objective function (Souza et al., 2021), the algorithm chooses random initial points to start the optimization. The surrogate model is key in BO because it encapsulates the beliefs about the objective function's shape. The most popular surrogate model is the GP, but others can be used, like Student-t Processes (Shah et al., 2014), or Bayesian Neural Networks (Li et al., 2024). The choice of the surrogate model is extremely problem dependent. In this study, we employ GP, but most derivations can easily be extended.

**Gaussian Process.** A GP (Rasmussen & Williams, 2006; Garnett, 2023) is a stochastic process, defined by an infinite collection of random variables, where any finite subset follows a multivariate normal distribution. GPs are particularly important in BO due to their compatibility with the Gaussian likelihood function. Consequently, after sampling points, the posterior distribution, as computed with Equation 2 remains a GP. A GP is fully characterized by its *mean* function, $\mu : \mathcal{X} \to \mathbb{R}$, and positive semidefinite covariance function or *kernel*, $k : \mathcal{X} \times \mathcal{X} \to \mathbb{R}$. A GP can then be expressed as $f \sim \mathcal{GP}(\mu, k)$. At initialization, the mean and kernel functions are specified to model a particular class of functions. Often, a non-informative mean function, such as $\mu \equiv 0$, is used. The choice of kernel is especially important, as it determines the spatial properties of the surrogate model, directly influencing its capacity to capture patterns in the data.

**Kernels.** Different kernels can be used to fit a GP on $f$ (Roman et al., 2019). In this study, we use the popular $5/2$-Matérn kernel (Chen et al., 2018; Rasmussen & Williams, 2006) defined as follows:

$$k(r) = \left(1 + \frac{\sqrt{5}r}{l} + \frac{5r^2}{3l^2}\right) \exp\left(-\frac{\sqrt{5}r}{l}\right) \tag{3}$$

where $r$ is the distance between two points of $\mathcal{X}$, $l$ is the positive lengthscale parameter.

**Posterior.** The posterior distribution refers to the updated probability distribution over $f$ after incorporating newly observed data. Considering Gaussian likelihood and noise in the observations, we have a closed form for the posterior distribution. As we collect samples and add them to the training dataset, the prior is updated to form the posterior distribution to improve the model's approximation of the objective function. Conjugated with the likelihood function - see Equation 2 - the posterior distribution $f|\mathcal{D}_{1:n}$ is a GP of mean $\tilde{\mu}$ and covariance $\tilde{k}$ (Kanagawa et al., 2018):

$$\tilde{\mu}(\mathbf{x}) = \mu(\mathbf{x}) + k_{\mathbf{X},\mathbf{x}}^T(\mathbf{K}_{\mathbf{X}} + \sigma_{noi}^2 I)^{-1}(\mathbf{y} - \boldsymbol{\mu}_{\mathbf{X}}) \tag{4}$$

$$\tilde{k}(\mathbf{x}, \mathbf{x}') = k(\mathbf{x}, \mathbf{x}') - k_{\mathbf{X},\mathbf{x}}^T(\mathbf{K}_{\mathbf{X}} + \sigma_{noi}^2 I)^{-1} k_{\mathbf{X},\mathbf{x}'} \tag{5}$$

In the above expression, $k_{\mathbf{X},\mathbf{x}'} = (k(\mathbf{x}_1, \mathbf{x}'), k(\mathbf{x}_2, \mathbf{x}'), ..., k(\mathbf{x}_n, \mathbf{x}'))^T$.

**Acquisition Function.** The acquisition function is very important in BO. This determines which new point will be collected. It provides a measure of utility for each new point to be sampled. The next point $\mathbf{x}^*$ is selected as the one which maximizes the acquisition function $AF$:

$$\mathbf{x}^* = \text{argmax}_{\mathbf{x} \in \mathcal{X}} AF(\mathbf{x}|\mathcal{D}_{1:t}) \tag{6}$$

There are several acquisition functions available (Wang et al., 2022), the earliest being Probability of Improvement (Kushner, 1964). A common choice is Expected Improvement (Bull, 2011), which is numerically stable. The one we use in this study is the popular Upper Confidence Bound (UCB):

$$AF(\mathbf{x}|\mathcal{D}_{1:t}) = \mu(\mathbf{x}) + \kappa\sigma(\mathbf{x}) \tag{7}$$

In the above expression, $\kappa > 0$ is a fixed *exploration-exploitation* trade-off hyperparameter and $\sigma(\mathbf{x}) = \sqrt{k(\mathbf{x}, \mathbf{x})}$. The algorithm tends to *exploit* areas where the potential reward is high (great values of $\mu$), or *explore* areas where the uncertainty of $f$ is high (great values of $\sigma$). This tradeoff is monitored by $\kappa$ which optimal value is problem-dependent.

## 2.3 RELATED WORK

Several studies have attempted to improve BO precision and computation time as the amount of data increases, but none fas addressed the issue by intervening in the query history, nor has offered strict guarantees of fixed computation time and a limit in memory usage.

**BO in High Dimension.** BO faces problems such as the curse of dimensionality, which leads to excessive computation time and memory usage as the number of training data points increases. An approach using Principal Component Analysis has been explored to improve scalability of BO in high-dimensional search spaces (Raponi et al., 2020), reducing CPU time by up to 10x, although the time still scaled with data complexity. Other hierarchical approaches have been proposed to address this issue, including specific applications like neurostimulation (Laferrière et al., 2020). However, while these methods reduce the number of full-dimensional queries by pre-training the GP in lower dimensions, they still require several iterations in the full-dimensional space.

**Domain Shrinking.** Numerous studies have attempted to improve BO by progressively reducing the search space to a confidence region. For example, TuRBO (Eriksson et al., 2020) is a method that optimizes GPs locally within multiple confidence regions, retaining only the best-performing regions in order to reduce the search space. By alternating local and global optimization phases, TREGO (Diouane et al., 2022) improves the efficiency of BO with Expected Improvement. We can also mention ZoMBI (Siemenn et al., 2023), an algorithm which limits the search space to the regions between the best points found, greatly reducing the optimization time. This algorithm works particularly well for "needle-in-a-haystack" problems where pruning of the input space is necessary.

**Sparse Gaussian Processes.** Another important approach to improve BO involves sparse Gaussian processes (SpGPs), an approximate version of standard GPs that uses a limited set of synthetic inducing points as a support set. This method relies on a fixed number of pseudo-entries to approximate the full GP (Snelson & Ghahramani, 2005), reducing computational costs while preserving accuracy. By optimizing these pseudo-entries, SpGPs capture relevant information from the dataset. Variational formulations further enhance this approach by optimizing the inducing inputs through maximizing a lower bound on the logarithmic marginal likelihood, allowing the inducing points to be optimized alongside the kernel parameters of the SpGP (Titsias, 2009). SpGPs have also been adapted to BO (McIntire et al., 2016), although their iterative training cost still scales with $n$, specifically at $\mathcal{O}(nm^2)$.

**Online Paging Algorithm.** The online paging problem is a classic memory management challenge, where memory is organized into a two-level structure: a fast memory cache of size $k$, and an unlimited slow memory. An adversary defines a sequence of requests to be processed by the paging algorithm. If a requested item is already in the cache, there is no associated cost. However, if the item resides in slow memory, it must be loaded into the cache at a fixed cost, requiring the eviction of one existing cache item to maintain the limit of $k$ elements. The eviction rule determines which item is removed from the cache in each round. Notably, it has been shown that employing a uniform random eviction rule can lead to a lower overall cost than any deterministic algorithm (Motwani & Raghavan, 1995). This holds even when the adversary is malicious and adapts the sequence of requests to exploit the paging algorithm's eviction strategy (Pruhs & Manber, 1991). These studies guided our research toward exploring random pruning strategies in BO.

## 3 MEMORY-PRUNING FOR BAYESIAN OPTIMIZATION (MP-BO)

Computing the exact posterior distribution (Equation 5) requires inverting and storing an $(n \times n)$ matrix, resulting in a computational complexity of $\mathcal{O}(n^3)$ and a storage requirement of $\mathcal{O}(n^2)$. In embedded systems, memory resources are often highly constrained, particularly when the goal is to minimize system size. By fixing the maximum number of data points $m \ll n$ to be retained throughout iterations, we can provide strict guarantees on computational and memory usage. This approach reduces the time complexity to $\mathcal{O}(m^3)$ and the storage requirements to $\mathcal{O}(m^2)$, where $m$ is fixed by design.

The approach adopted in this study is as follows: once the designed resource limit is reached, we continue to perform BO optimization, updating a full GP distribution, *i.e.*, with no domain shrinking. However, at each query, collected points from the training set are removed iteratively, thereby strictly limiting the dimension of the matrix to be inverted. This size limit provides precise guarantees of computational cost, both in time and in memory, for each future iteration. Consequently, the optimization is performed on a subset of all collected samples rather than on the entire dataset. The challenge lies in selecting which points to prune while maintaining a model that accurately captures the desired $f$-optimum. Algorithm 1 outlines our approach, MP-BO, where $q^*$ represents the iteration at which we begin to remove training points.

---

**Algorithm 1** Bayesian Optimization with Memory Pruning (MP-BO)

1: **Init:** Randomly sample a point $\mathbf{x}_1$ and its response $y_1$.
2: $\mathcal{D}_{1:1} := \{(\mathbf{x}_1, y_1)\}$
3: Set $\mu(.) = 0, \sigma(.) = \sqrt{k(.,.)}$
4:
5: **for** $n = 1, 2, \ldots$ **do**
6:      $\mathbf{x}_{n+1} = \arg\max_{\mathbf{x}} AF(\mathbf{x}|\mathcal{D}_{1:n})$                 ▷ Find new $\mathbf{x}_{n+1}$ to sample
7:      $y_{n+1} = f(\mathbf{x}_{n+1}) + \varepsilon_{n+1}$                    ▷ Sample the objective function
8:      **if** $n \geq q^*$ **then**
9:          $(\tilde{\mathbf{x}}, \tilde{y}) = u(\mathcal{D}_{1:n})$             ▷ Find a query to delete and remove it from training set
10:          $\mathcal{D}_{1:n} = \mathcal{D}_{1:n} \setminus \{(\tilde{\mathbf{x}}, \tilde{y})\}$
11:      **end if**
12:      $\mathcal{D}_{1:n+1} = \mathcal{D}_{1:n} \cup \{(\mathbf{x}_{n+1}, y_{n+1})\}$        ▷ Augment the data set and update the surrogate model
13:      Compute $\mu$ and $\tilde{\sigma}$ of the GP
14:      **for all** $\mathbf{x} \in \mathcal{X}$ **do**
15:          $\sigma(\mathbf{x}) = \min(\sigma(\mathbf{x}), \tilde{\sigma}(\mathbf{x}))$          ▷ Keep the minimal uncertainty $\sigma(\mathbf{x})$ for each $\mathbf{x} \in \mathcal{X}$
16:      **end for**
17: **end for**

---

When removing a data point, the GP uncertainty $\sigma(\mathbf{x})$ will be raised. This might mislead the acquisition function UCB, which usually depends on $\sigma(\mathbf{x})$, to sample again the pruned data point. To avoid this phenomenon, we have established the rule $\sigma(\mathbf{x}) = \min(\sigma(\mathbf{x}), \tilde{\sigma}(\mathbf{x}))$, which forces the GP uncertainty to follow the minimum between $\sigma(\mathbf{x})$, calculated at the previous query, and $\tilde{\sigma}(\mathbf{x})$), calculated after pruning and adding a new data point.

### 3.1 MEMORY PRUNING STRATEGY

We are looking for an efficient strategy to select a past data point and then remove it from the training set at each iteration. Figure 1 shows that our strategy described in Algorithm 1 largely reduces, and maintains constant, the computation time per each future query.

This algorithm is not designed to achieve better performance than VANILLA BO. Rather, our aim is to minimize performance loss while providing strict guarantees on a limit in computation cost for any given query. In this context, a theoretical idea would be to select the query which minimizes the difference between the updated posterior distribution containing this point and the updated distribution without this point (Titsias, 2009). One could use the Kullback-Leibler Divergence (Belov & Armstrong, 2011) to get an idea of which query to delete. However, calculating the posterior distribution can be particularly costly. For this reason, we explore different strategies to only compute the posterior distribution once, but with the best collected points. We observed that this strategy is performant for GP-BO, even in large search spaces. At each iteration, we remove a random already

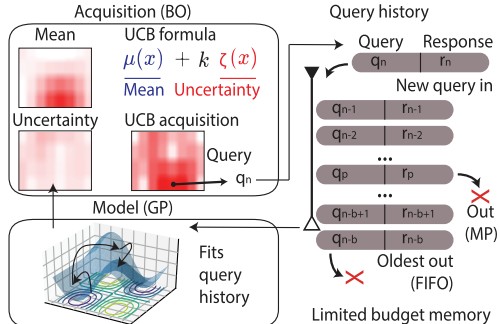 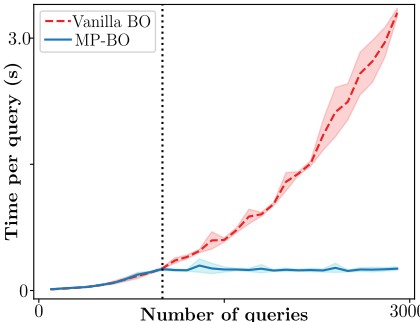

Figure 1: **Schema of MP-BO iteration and computing time.** In the schema, two possible pruning choices are depicted: the first-in-first-out (FiFo) approach, and our MP-BO algorithm. On the right, the evolution of computing time required for one iteration (time spent in the optimization part and in the update part, where the mean and kernel functions of the GP are updated) is plotted against the number of queries. We see that the computational time becomes constant after applying MP-BO. Data are presented as mean $\pm$ standard error of mean (SEM). $q^* = 1000$ in this experiment, indicated by the vertical dashed line.

sampled point from the training set, with the exception of the latest acquisition and the current best point, in order to protect continual learning and optimization. We also explored alternative strategies, inspired by the online paging problem and by studies on the impact of outliers (Liu et al., 2020; Siemenn et al., 2023; Martinez-Cantin et al., 2017), but these alternatives did not perform better than random drop. Insights and results can be found in Appendix B.

While MP-BO provides strict guarantees on computation time and randomized eviction proves to be more robust than other deterministic methods, there is no guarantee that this pruning approach is optimal. Depending on the problem, it may benefit from design-specific tuning. For instance, in highly time-varying optimization problems, a designer may prefer to bias randomized eviction toward older data points to better follow temporal changes. In this work, we focus on stationary problems and demonstrate the versatility of MP-BO with randomized eviction exclusively.

## 4 EXPERIMENTS

### 4.1 EXPERIMENTAL SETUP

**Datasets.** To assess the performance of our algorithm, we evaluate it on three different benchmarks of classic optimization problems: Ackley, Michalewicz and Hartmann (Surjanovic & Bingham, 2013), see Appendix A.1. We consider our datasets as discretized because it is relevant for real world applications, especially in embedded systems. Then, we applie our algorithm on a neurostimulation dataset obtained on non-human primates (Bonizzato et al., 2021; 2023), involving electromyographic (muscle) responses measured when an electrical microstimulation is applied in the brain motor cortex. The bi-dimensional location of the stimulation is optimized to find the strongest evoked movement.

**Baseline and Evaluation Metrics.** In this study, we compare our algorithm with the VANILLA BO algorithm. We use a measure of regret to assess the performance of both algorithms. We also compare our results with different pruning strategies like FiFo or different types of centrality estimators, with results in Appendix B.

**Regret:** Let $\mathbf{x}^*$ be a maximizer of $f$, i.e. $\mathbf{x}^* = \text{argmax}_{\mathbf{x} \in \mathcal{X}} f(\mathbf{x})$ and suppose that at iteration $n \in \mathbb{N}$ in the algorithm, we predict $\mathbf{x}_n$ as the best point. Then, the *instantaneous regret* $r_n$ is defined by $r_n := f(\mathbf{x}^*) - f(\mathbf{x}_n)$. The instantaneous regret shows if the algorithm converges and if so, how fast does it converge. Our objective is to minimize its value. Theoretical bounds for the cumulative regret with the UCB acquisition function already exist (Srinivas et al., 2012).

**Implementation Details and Hyperparameters.** In our experiments, we perform our tests on 30 independent repetitions. We consider discretized datasets, where the discretization steps are described in Appendix A.1. We preprocess our data with a min-max normalization such that all the observations are between 0 and 1. Moreover, we use the `gpytorch` framework (Gardner et al., 2018), which allows us to optimize the lengthscale and noise parameters of the GP using maximum a posteriori estimation (MAP). As we use the UCB acquisition function, we need to determine the *exploration-exploitation* trade-off hyperparameter $\kappa$. To do so, we run the algorithm with several values for the hyperparameter and then use the one which gives the best regret. We perform the $\kappa$ optimization on the VANILLA BO algorithm and use the same value for MP-BO. Thus, we obtain a conservative setting, where $\kappa$ is ideal for VANILLA BO, but has not been tuned for MP-BO. We set the observation noise hyperparameter to 0.025.

Moreover, we update dynamically the GP's variance $\sigma(\mathbf{x}) = \sqrt{k(\mathbf{x}, \mathbf{x})}$ by only keeping the minimal value between $\sigma(\mathbf{x})_{t-1}$ and $\sigma(\mathbf{x})_t$ for each $\mathbf{x} \in \mathcal{X}$. This is a very important step in our algorithm, as doing so helps the algorithm to converge and avoids to overly revisit the points that have just been dropped from memory.

Finally, we define $q^*$, the iteration at which we start to apply the memory-pruning strategy. Most of this study uses a value of $q^* = 20$, unless indicated otherwise.

**Hardware Configuration.** The experiments are conducted on a MacBook Pro with an Apple M1 chip, featuring 8 cores (4 performance cores and 4 efficiency cores) and 8 GB of unified memory, running macOS. This setup represents a conservative choice when compared to use cases involving more compact and embedded systems, where stricter limits on computational power exacerbate the issue of unrestrained growth in execution time.

## 4.2 EXPERIMENTAL RESULTS ON SYNTHETIC DATASETS

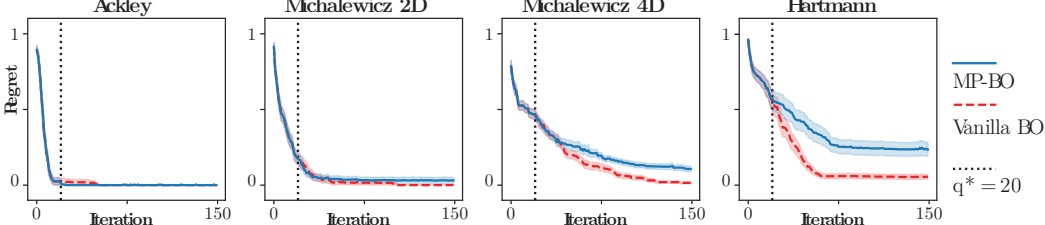

Figure 2: **Regret on different optimization benchmarks.** Data: mean $\pm$ SEM over all the repetitions.

First, we evaluate MP-BO performance on the following synthetic datasets: Ackley, Michalewicz and Hartmann (Surjanovic & Bingham, 2013), see Appendix A.1. In this study, we compare MP-BO and VANILLA BO with a fixed value for $q^* = 20$. Results are displayed on Figure 2.

We can see that MP-BO achieves robust performance, even if it is slower to converge. Importantly, it always displays continued learning after $q^*$. Performance is highly dependent on the number of training data we allow the algorithm to store, thus on the hyperparameter $q^*$. There is a clear trade-off between the number of observations to maintain and the time and memory complexity. Setting a very low $q^*$ will surely reduce the computational cost of the algorithm, but will need a lot of iterations to converge.

**Hyperparameter $q^*$.** We ask how much we can reduce the memory usage in MP-BO without significantly compromising performance. Specifically, we seek to understand whether MP-BO can continue learning the representation of the objective function after fixing the amount of query history used for training. To evaluate this, we compare the final performance of MP-BO with that of VANILLA BO at iteration $q^*$. The difference in performance indicates whether MP-BO continues to learn effectively acquiring the new $q_{final} - q^*$ training points. The results of this experiment are presented in Figure3. Applying our strategy does not prevent MP-BO from learning the represen-

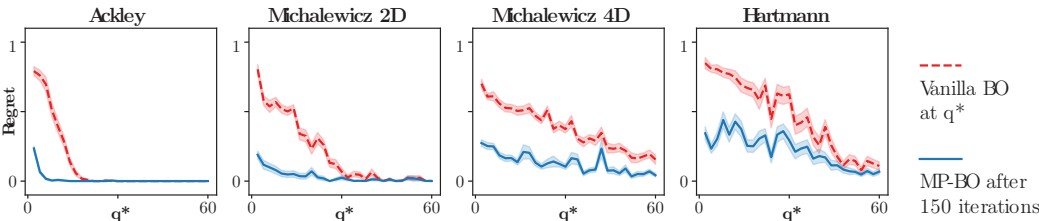

Figure 3: **Final Regret after 150 iterations compared to VANILLA BO's regret at** $q^*$**.** In this figure the impact of $q^*$ on MP-GPBO is shown. In most cases, a too low value for $q^*$ prevents our algorithm to converge as it needs a minimal number of training points for optimization. MP-BO continues to learn after $q^*$. Data: mean $\pm$ SEM.

tation of the objective function. Although MP-BO exhibits the highest learning delta at lower $q^*$ values, these values are also associated with incomplete learning, meaning that the final performance of MP-BO differs from what would be achieved with a larger $q^*$. In many cases, intermediate $q^*$ values strike a balance, delivering both robust final performance and significant learning gains.

**Robustness and Consistency.** We study the robustness of MP-BO in noisy datasets or large input spaces. Indeed, since we do not use continuous input spaces but discretized ones, the performance can be impacted by the grid size we choose. We thus compare the performance on the 2-dimensional Ackley function and increasing the number of available discrete input points. Knowing that BO can suffer from the curse of dimensionality (Papenmeier et al., 2022) and have trouble converging in higher-dimensional datasets, we also increase the dimension of the Ackley function and assess the performance of VANILLA BO and MP-BO. Results are shown in Figure 4. The experiment shows that MP-BO does not particularly suffer from an increasing input dimension, provided that a reasonable amount of learning has already occurred at $q*$.

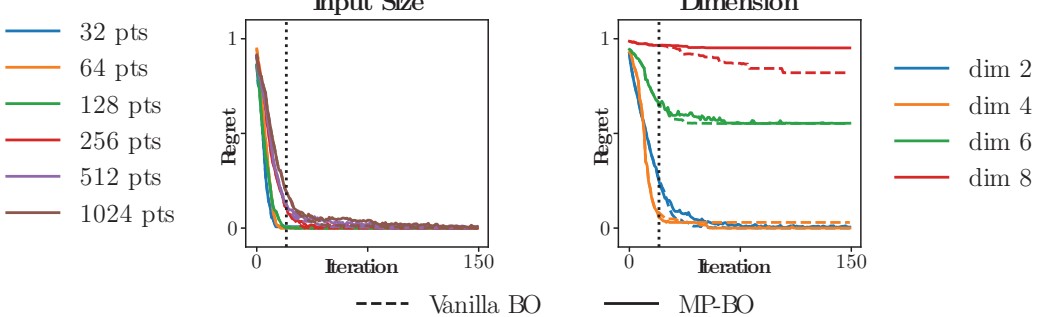

Figure 4: **MP-BO performance for diverse problem dimensions.** On the left, we use the 2-dimensional Ackley function, with a varying number of available discrete input points. On the right, we used a fixed number of discrete input points with a varying number of dimensions. VANILLA BO performance is represented by the dashed lines, while MP-BO is represented by the solid lines. We do not display standard errors for visibility purposes. The vertical dotted line represents $q^* = 20$.

We then turn our attention to the observation noise. Since we do not have direct access to the objective function, we only observe values corrupted by noise: $\mathbf{y}_i = f(\mathbf{x}_i) + \epsilon_i$, with $\epsilon_i \sim \mathcal{N}(0, \sigma_n^2)$. We evaluate the performance of MP-BO under varying levels of noise, ranging from $0\%$ to $50\%$ of the optimal value. Figure 5 presents representative results for noise levels of $0\%$, $2.5\%$, and $20\%$, for brevity. The figure shows that for small datasets like Ackley and Michalewicz 2D, the noise level has little impact. However, for larger datasets, optimization becomes more challenging with VANILLA BO, and the effect of noise on MP-BO becomes more pronounced.

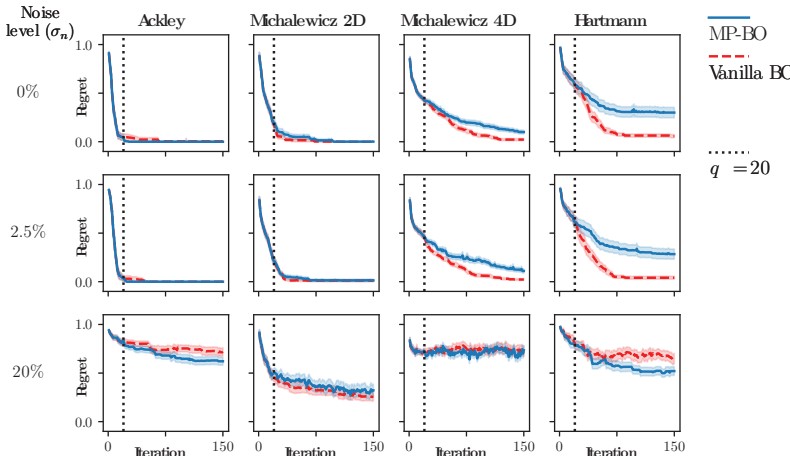

Figure 5: **Influence of the noise level on the performance of MP-BO.** Data: mean $\pm$ SEM.

**Time Reduction.** As seen in the previous results, MP-BO manages to continue learning the objective function representation after $q^*$. Setting a too low value for $q^*$ makes it very slow to converge, but since it is faster than VANILLA BO, we can afford to perform more iterations. In Figure 6, we compare VANILLA BO and MP-BO, for the same duration, to determine the regret each algorithm can achieve when considering, more meaningfully, the total execution time, as opposed to the number of queries. MP-BO being faster, it can perform more iterations and thus reach a smaller value of regret in the same amount of time. Thus, MP-BO can have very interesting applications when onboarded in systems with limited computing power, where the computation time at each iteration would otherwise rapidly rise beyond the constraints of the optimization problem.

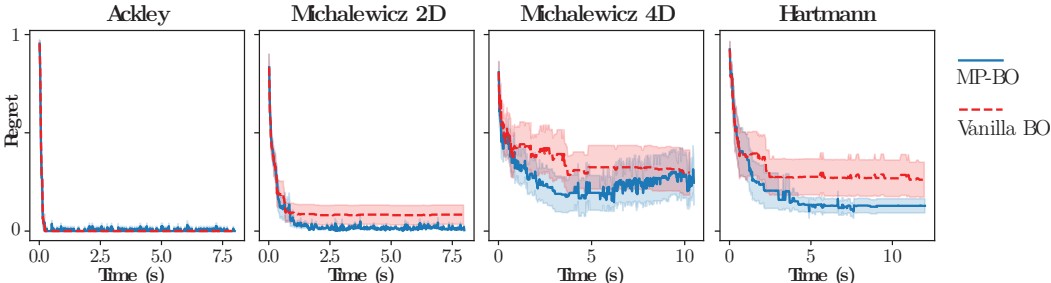

Figure 6: **Regret with respect to the time spent in the optimization.** The x-axis represents the total time spent in optimization through iterations. The value of $q^*$ is 20 and we use a noise of 10%. VANILLA BO is run on 300 queries, while MP-BO can perform approximately 550 queries in the same timeframe. Both are repeated 10 times. Data: mean $\pm$ SEM.

### 4.3 RESULTS ON REAL WORLD DATASETS

Multiple domains can benefit from a faster optimization process with guarantees of execution time, and here we present a real-world example on neurostimulation optimization. We utilize a dataset collected in non-human primates (Bonizzato et al., 2021; 2023), with the goal of selecting the optimal brain stimulation pattern that maximizes muscle responses in a 2-dimensional input space. The responses are noisy, so each stimulation option is sampled multiple times to estimate the average response, which is then considered the ground truth (Figure 7).

In this problem, muscle responses are collected within 100 ms of stimulation, theoretically allowing a high rate of optimization query iterations. However, the execution time per query for BO would increase rapidly and continuously over time, ultimately limiting the achievable repetition rate.

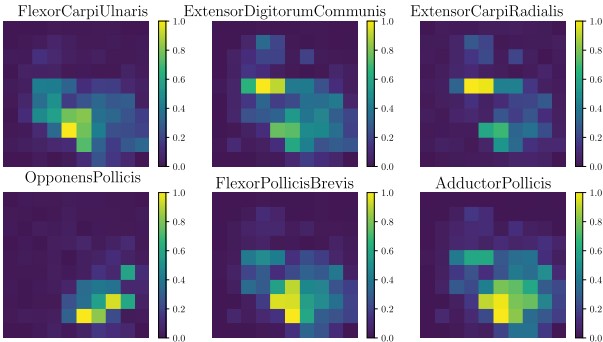 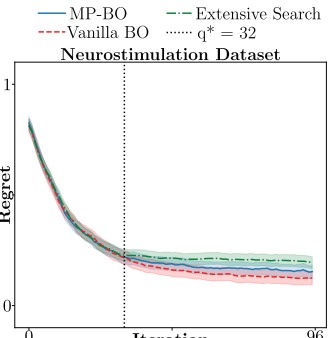

Figure 7: **MP-BO performance on a neurostimulation dataset.** On the left, average muscle responses to cortical stimulation are displayed for each input option. This is the ground truth of the function to optimize. The input space is a $(10 \times 10)$ grid. On the right, the average result of BO on 4 non-human primates. Our experiment is done on 20 repetitions per subject. Data: mean $\pm$ SEM.

In Figure 7, we show the result of optimization on twenty-two EMGs from four non-human primates. Here, we show another benchmark optimization method, called Extensive Search. This is the base method used by human operators to determine the optimal input in neuroscience research practice (Bonizzato et al., 2023) and corresponds to sampling all input points in random order. We compare MP-BO with applying Extensive Search after $q*$. This benchmark is relevant due to its minimal computational cost; however, it suffers from more significant performance degradation compared to MP-BO.

This experiment provides an empirical demonstration of MP-BO in solving an engineering problem where practical solutions are scarce (Bonizzato et al., 2023). Given that the neural interface is implanted, its optimization must rely on limited computational resources, making MP-BO well suited for such scenarios.

## 5 CONCLUSION

We developed a new method to adapt BO to a context where memory and/or time are limited. Our Memory-Pruning algorithm is capable of learning and predicting the objective function's maximum. Furthermore, it has strict guarantees on capping execution time to a desired value.

One limitation is that although MP-BO seems to be able to find the maximum of the objective function, the number of iterations required may be large, and there is currently no guarantee that the algorithm will converge. Further developments are needed to demonstrate convergence, if indeed convergence occurs. We believe that our technique can have a real impact on the performance of embedded BO systems, particularly in embedded systems for autonomous neurostimulation.

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

# A  DATASETS USED

## A.1  BENCHMARK DATASETS

All test functions are sourced from a library of optimization functions (Surjanovic & Bingham, 2013), and the evaluations of the true optimum values are drawn from (Vanaret et al., 2020). We discretize the input space and apply our strategy to these datasets.

Table 1: Test function and their domain.

| Function name | Dimension | Size | Test region |
|---|---|---|---|
| Ackley | 2 | $64 \times 64$ | $[-32, 32]^2$ |
| Hartmann | 6 | $5^6$ | $[0, 1]^6$ |
| Michalewicz | $2, 4$ | $64 \times 64, 10^4$ | $[0, \pi]^2, [0, \pi]^4$ |

### A.1.1  ACKLEY

$$f(\mathbf{x}) = -a \exp\left(-b\sqrt{\frac{1}{d}\sum_{i=1}^{d} x_i^2}\right) - \exp\left(\frac{1}{d}\sum_{i=1}^{d}\cos(cx_i)\right) + a + \exp(1) \tag{8}$$

Where $a = 20$, $b = 0.2$, $c = 2\pi$ are the usual parameters values and $d$ is the dimension of the input space.

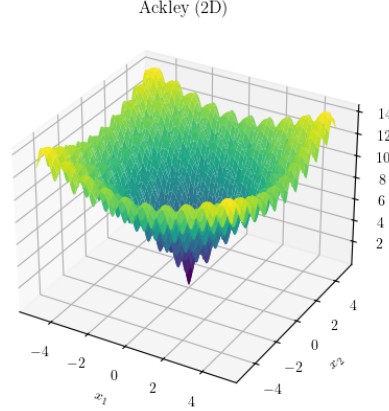

Figure 8: Ackley function in two dimensions. Since our problem involves maximization, we optimize the negative of the function.

### A.1.2  MICHALEWICZ

$$f(\mathbf{x}) = -\sum_{i=1}^{d} \sin(x_i)\sin^{2m}\left(\frac{ix_i^2}{\pi}\right) \tag{9}$$

Where $m = 10$ is the usual value and $d$ is the dimension.

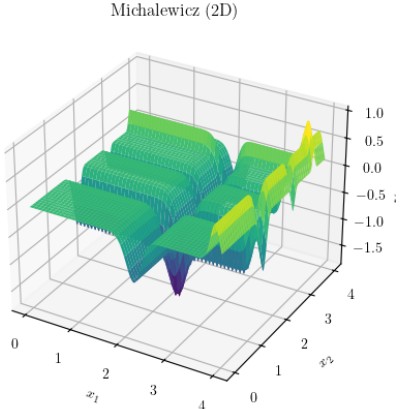

Figure 9: Michalewicz function in 2 dimensions. Since our problem involves maximization, we optimize the negative of the function.

### A.1.3 HARTMANN

$$f(\mathbf{x}) = -\sum_{i=1}^{4} c_i \exp\left(-\sum_{j=1}^{6} A_{ij}(x_j - P_{ij})^2\right) \tag{10}$$

Where:

$$c = [1.0, 1.2, 3.0, 3.2]$$

$$A = \begin{bmatrix} 10 & 3 & 17 & 3.5 & 1.7 & 8 \\ 0.05 & 10 & 17 & 0.1 & 8 & 14 \\ 3 & 3.5 & 1.7 & 10 & 17 & 8 \\ 17 & 8 & 0.05 & 10 & 0.1 & 14 \end{bmatrix}$$

$$P = \begin{bmatrix} 0.1312 & 0.1696 & 0.5569 & 0.0124 & 0.8283 & 0.5886 \\ 0.2329 & 0.4135 & 0.8307 & 0.3736 & 0.1004 & 0.9991 \\ 0.2348 & 0.1451 & 0.3522 & 0.2883 & 0.3047 & 0.6650 \\ 0.4047 & 0.8828 & 0.8732 & 0.5743 & 0.1091 & 0.0381 \end{bmatrix}$$

# B  STRATEGIES

In addition to our tests with MP-BO, we evaluate multiple other non-random, deterministic strategies for removing a sampled training point. These include a FiFo approach, which eliminates the oldest query; an approach that removes the query with the worst response; and two approaches that target intermediate queries, selected by the arithmetic and geometric mean responses, respectively. These are reported in Table 2. In Figure 10, we show that the optimization performance achieved for other strategies does not exceed that of MP-BO.

Table 2: Alternative pruning strategies.

| Strategy | Index in $\mathcal{D}_{1:t}$ selected |
|---|---|
| MP-BO | $\mathcal{U}(1,t) \setminus \arg\max_i \{y_i\}$ |
| FiFo | $1$ |
| Worst | $\arg\min_i \{y_i\}$ |
| Mean | $\arg\min_i \{|\frac{1}{t} \sum_{j=1}^{t} y_j - y_i|\}$ |
| GeoMean (He et al., 2019) | $\arg\min_i \{|\sqrt[t]{\prod_{j=1}^{t} y_j} - y_i|\}$ |

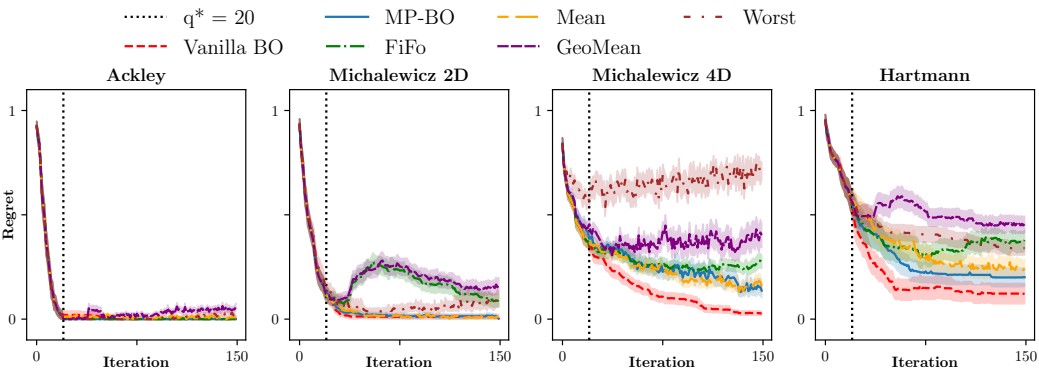

Figure 10: **Regret comparison of differents pruning strategies with VANILLA BO.**

