# OpenReview forum: "Memory-Pruning Algorithm for Bayesian Optimization with Strict Computational Cost Guarantees"
_ICLR.cc/2025/Conference — ICLR 2025 Conference Withdrawn Submission_

### Official Review · Reviewer_kgBB · 2024-10-21

**Soundness:** 2
**Presentation:** 2
**Contribution:** 2
**Rating:** 3
**Confidence:** 4

**Summary:**

This paper explores an extension of the GP-UCB algorithm for optimizing cheap-to-query black-box functions in discrete domains. By pruning the memory of observed points in the exact GP model, the proposed approach achieves faster computation at the cost of some performance. However, the empirical results show that the performance degradation is minimal in the tasks the authors tested. This algorithm proves useful when the objective function is inexpensive to query, there is an upper bound on the computational overhead for selecting the next query—particularly in scenarios where time is plentiful but monetary cost is high, and longer iteration cycles are needed (otherwise this algorithm reduces to standard UCB).

**Strengths:**

- Explicitly upper-bounding the computation time is an important problem for optimizers in general, though Bayesian optimization may not be the first choice in such scenarios.
- The neurostimulation example is interesting and effectively highlights the context of limited computation time. However, the need for longer iteration cycles is not entirely clear, which raises questions about the importance of sample efficiency in this case.
- The core idea is simple and easy to follow. However, the assumptions are dispersed throughout the text, and it is only later in the paper that we learn this method is limited to discrete domains.

**Weaknesses:**

- **This algorithm is not a global optimizer**.
It lacks the no-regret property that guarantees GP-UCB as a global optimizer. As shown in [1], the no-regret property is defined as $\lim_{T \rightarrow \infty} \frac{R_T}{T}= 0$, where $R_T := \sum_{t =1}^T r_t$ is the cumulative regret, $r_t$ is an instantaneous regret, and $t$ is the iteration step. This no-regret property assures that original GP-UCB algorithm is a global optimizer. However, this algorithm restricts the number of queries, i.e., $\lim_{T\rightarrow \infty} T = m$, meaning the regret never converges to zero. Intuitively, the GP uncertainty $\sigma$ is submodular with respect to the number of data points, allowing it to converge asymptotically for infinitely many queries. Unfortunately, this algorithm will not achieve no-regret. Therefore, the authors should clarify that this is a heuristic, similar to TurBO, which explicitly indicated this in the title (“local optimization”). The descriptions in line 323 are misleading in this regard.
- **Inaccurate complexity analysis**.
While the complexity analysis is correct for a single point $x_t \in \mathcal{X}$, in practice, the algorithm adopts a memorization step for $\sigma_t(\mathcal{X}), \sigma_{t-1}(\mathcal{X}) $, which requires
$N = |\mathcal{X}|$ in a discrete domain. As a result, the total complexity is $\mathcal{O}(N m^3)$ for time and $\mathcal{O}(m^2 + 2 Nm)$ for memory. This makes the complexity quite comparable to that of sparse GPs. Since sparse GPs do not need to scan the entire domain, their complexity is $\mathcal{O}(M n m^2)$ for time, where $M$ is the number of iterations in the acquisition function maximizer (e.g., L-BFGS-B), and $m$ is number of inducing points. When $M \ll N$, sparse GPs are faster. Therefore, this new algorithm is not necessarily always superior to sparse GPs, and a more reasonable comparison between the two should be provided. (Sparse GP can also limit the computation time by setting appropriate $m$ and iteration $T$).
- **Why Bayesian optimization?**
Bayesian optimization may not be the first choice for optimizing cheap-to-query functions. As discussed earlier, this algorithm is not a global optimizer, but rather a heuristic approach. There are numerous other efficient heuristics for sample-efficient black-box optimization, such as CMA-ES, NES, and others, which should be considered for comparison. Additionally, if computational time is a significant factor, parallel computation (i.e., batch BO) could be a viable alternative. In environments where costs are low, cloud computing—similar to how ChatGPT operates—is also an option. As a result, the motivation for using this algorithm in such scenarios remains unclear.
- **The better baseline that is not included in the paper**.
As the authors state, the goal is to balance computation time and convergence rate. However, this method seems to sacrifice the convergence rate too heavily, particularly since it loses the no-regret property. For example, [2] introduced a more principled approach to memory-pruning in GP-UCB, which includes regret analysis. This method achieves a similar computational time complexity of $\mathcal{O}(m^3)$ while still maintaining the no-regret property. Similarly, sparse GPs may also perform reasonably well. When the upper bound of computational overhead is known, we can use reverse calculation to determine the number of inducing points $m$, ensuring a better balance between computational efficiency and convergence.
- **Minor points.** Kappa is not fixed in GP-UCB. This should increase with iterations. Read [1] carefully.
- **Citations**
- [1] Niranjan Srinivas, Andreas Krause, Sham M. Kakade, Matthias Seeger, Gaussian Process Optimization in the Bandit Setting: No Regret and Experimental Design, ICML 2010
- [2] Daniele Calandriello, Luigi Carratino, Alessandro Lazaric, Michal Valko, Lorenzo Rosasco, Scaling Gaussian Process Optimization by Evaluating a Few Unique Candidates Multiple Times, ICML 2022

**Questions:**

The questions in the above weakness section.

---

> ### Author Response · Authors · 2024-11-19
>
> We would like to thank you all for spending time reading and providing feedback on our paper. In the following, we wish to extend our responses to the points you raised. No matter where this will lead the paper, we sought not to leave your comments and suggestions unanswered.
>
> 1. This algorithm is not a global optimizer. It lacks the no-regret property that guarantees GP-UCB as a global optimizer.
>
> This is absolutely true and we regret that this came through by lack of clarity. Any future version of this manuscript should contain this consideration. As you noted, it is possible to prove that we cannot guarantee convergence to the global maximum.
>
> 2.While the complexity analysis is correct for a single point xt in X, in practice, the algorithm adopts a memorization step for N= |X| in a discrete domain. As a result, the total complexity is bigO(Nm^3) for time and bigO(m^2+2Nm) in memory.
>
> We believe the correct total complexity including N would be bigO(N+m^3) for time and bigO(m^2+N) in memory. Sparse GPs 1) always contain the n term in the complexity, 2) are bound to have execution time -> inf for large q. Whereas MP-BO is strictly capped in execution time.
> It’s true there are plenty of problems where Sparse GP is to be preferred. However, here we tackle the set of problems where one has strict constraints on execution time, this is the only GPBO method to provide such guarantees, and, as such, a contribution that we believe important.
>
> 3. Why Bayesian optimization? There are numerous other efficient heuristics for sample-efficient black-box optimization, such as CMA-ES, NES, and others, which should be considered for comparison.
>
> We agree that model-free blackbox optimization algorithms can be a choice. These algorithms lack some of the best properties of GPBO, like propagating knowledge to neighbors, and explicit weighting of exploitation and uncertainty. Here we sought to offer an option for this. However, you are right, our paper would improve with some model-free options that also have bounded execution time in our benchmark.
>
> 4. If computational time is a significant factor, parallel computation (i.e., batch BO) could be a viable alternative. In environments where costs are low, cloud computing—similar to how ChatGPT operates—is also an option. As a result, the motivation for using this algorithm in such scenarios remains unclear.
>
> Unfortunately, none of these methods give strict guarantees in execution time (i.e., constant capped time to execute), which is a desirable factor in applications such as robotics, finance, and neuromodulation.
>
> 5. The better baseline that is not included in the paper. This method seems to sacrifice the convergence rate too heavily, particularly since it loses the no-regret property. For example, [2] introduced a more principled approach to memory-pruning in GP-UCB, which includes regret analysis. This method achieves a similar computational time complexity of bigO(m^3) while still maintaining the no-regret property. Similarly, sparse GPs may also perform reasonably well. When the upper bound of computational overhead is known, we can use reverse calculation to determine the number of inducing points, ensuring a better balance between computational efficiency and convergence.
>
> What you are suggesting with [2] is for any given q*, a particular setting of GP-UCB can be deployed, that is associated with guarantees on regret. [2] does not continue searching after that point.  The approach to maximizing f is “to first choose points xt so as to estimate the function globally well, then play the maximum point of our estimate.” This means that no device is in place to reduce computation time after the forced maximum number of queries. We agree that we could add this comparison as a benchmark, but we propose that this method + MP would be more successful, as it can keep learning after the t chosen points. Figure 3 shows continual learning after q*. In fact, we think that combining your suggestion of using [2] until q* and then continuing with MP would likely result in the most performant optimization strategy.
> As reported above, we also agree that sparse GP can perform really well in a large set of problems, but when hitting the computation cap, pruning should commence. As such, one could also consider Sparse GP + MP.
> In short, we propose MP as something one should do after hitting the cap in computation time, independently from the method used before the limit.
> Finally, we are completely on board with the idea that there are many interesting trade-offs between computational efficiency and convergence. Here we are trying to tackle the extreme one: computational time guarantees.
>
> 6. Kappa is not fixed in GP-UCB.
> This is also true. One may argue that the vast majority of GP-UCB implementations in literature and applied science use fixed kappas, but a better choice would give a dynamic value for it (at the cost of introducing one more hyperparameter in kappa change rate).

---

> > ### Comment · Reviewer_kgBB · 2024-11-25
> >
> > Thank you for your response. I will maintain my score.

---

### Official Review · Reviewer_GHdB · 2024-10-30

**Soundness:** 2
**Presentation:** 3
**Contribution:** 2
**Rating:** 3
**Confidence:** 3

**Summary:**

To address the computational and memory challenges of Bayesian optimization (BO) on large datasets, this article introduces a Bayesian optimization algorithm with memory pruning (MP-BO). MP-BO limits the maximum size of the training data by selectively updating the data used for the proxy model. When the dataset exceeds a predefined limit, new queries are incorporated while older data points are removed from the training set. The algorithm’s performance was validated across various aspects on both synthetic and real-world datasets, demonstrating its effectiveness.

**Strengths:**

Completed work: The article is clearly written, and the charts used for the presentation are visually appealing.
Clear motivation: The memory pruning strategy could be valuable in memory-constrained scenarios.

**Weaknesses:**

Weakness:
Unclear notation: In the whole paper, sometimes mathematic notation happens without explanation which hinder the understanding of readers seriously. For example: “m” in the abstract, author just treat it as a “fixed value” but did not make it clear what value it is. Also in the algorithm, the author write u without any further explanation.

Insufficient illustrations: According to my understanding of MP-BO, the most important point is how to find the point to be deleted, but in the paper, the author just mentioned use KL-divergence to decided but doesn’t provide any mathematical equations to explain how to use the KL-divergence.

Also, as mentioned in the first section, the author says “m” is a fixed value, but how to decide the value of m? And same for q^*

Baseline implementation: There are too few empirical comparison methods, only with Vanilla BO. There is plenty of literature on BO, I wonder whether the author consider comparing MP-BO with other BO methods?

**Questions:**

What is the specific methodology for applying KL-divergence in this context?

How is the value of  $q^*$  determined?

Is the experimental time for the Michalewicz 4D and Hartmann problems presented in Figure 6 insufficient? The results suggest that neither method has fully converged.

What considerations are there for high-dimensional outputs and alternative acquisition functions?

---

> ### Author Response · Authors · 2024-11-19
>
> We would like to thank you all for spending time reading and providing feedback on our paper. In the following, we wish to extend our responses to the points you raised. No matter where this will lead the paper, we sought not to leave your comments and suggestions unanswered.
>
> 1. In the whole paper, sometimes mathematic notation happens without explanation
>
> We apologize for the bad notation. We certainly want to revise this.
>
> 2. in the paper, the author just mentioned use KL-divergence to decided but doesn’t provide any mathematical equations to explain how to use the KL-divergence.
>
> Sorry for the lack of clarity. KL-divergence was incidentally proposed as a possible option which was discarded because of the computing cost. Queries are here dropped at random to achieve a fixed time of execution.
>
> 3. “m” is a fixed value, but how to decide the value of m? And same for q^*
>
> “m” and “q*” are de facto interchangeable in our text, which we agree it’s not great. We should have used q* everywhere. Thank you also for the suggestion of giving a guideline for selecting q*, which is very simple: with just one preliminary testing with random values, anyone implementing this can determine for increasing qi what’s the maximum value q* that is associated with execution time strictly inferior to t_max, where t_max is the maximum allowable cycle time.
>
> 4. There are too few empirical comparison methods, only with Vanilla BO. consider comparing MP-BO with other BO methods?
>
> We argue that no other GP-based optimization method can work on this problem within a fixed guarantee of execution time. However, alternative model-free methods do have fixed execution time and could have been tested here, and in a future version of this paper we could include those.

---

### Official Review · Reviewer_KoAo · 2024-11-01

**Soundness:** 2
**Presentation:** 2
**Contribution:** 2
**Rating:** 3
**Confidence:** 4

**Summary:**

This paper proposes a memory-pruning Bayesian optimization method that improves the running speed of BO.
The idea is discard a subset of the training data so that the training data size stays constant during Bayesian optimization.
Concretely, the author propose randomly deleting a training data after adding each new data point to the training set after certain iterations.
The authors verify this strategy improves the running speed and does not degrade the BO performance too much compared to vanilla BO.

**Strengths:**

- The proposed method is simple, intuitive, and easy to implement.

**Weaknesses:**

- Motivation is unclear / weak experiments.
While I agree BO is costly when the number of training data is large, all experiments in the paper have at most 200 data points (including the real-world experiments).
Training data of these sizes can be handled trivially by modern computers.
Thus, I am not sure if there is a need dropping training data on these problems.

- Lack of baselines.
The authors propose reducing the data size by dropping training data randomly.
However, there is no other baselines to access how good or how bad this idea is.
Clearly, randomly dropping training data is one of the easiest heuristics that one can come up with.
At this stage, I am not sure if this paper reveal any interesting insights of the problem that the authors are trying to solve.

**Questions:**

In Line 15 of Algorithm 1, how enumerate all \\(x \in \mathcal{X}\\)? Is the domain discretized?

---

> ### Author Response · Authors · 2024-11-19
>
> We would like to thank you all for spending time reading and providing feedback on our paper. In the following, we wish to extend our responses to the points you raised. No matter where this will lead the paper, we sought not to leave your comments and suggestions unanswered.
>
> 1. all experiments in the paper have at most 200 data points (including the real-world experiments). Training data of these sizes can be handled trivially by modern computers. Thus, I am not sure if there is a need dropping training data on these problems.
>
> The rationale behind this choice was to focus on a key use case: addressing a number of queries sufficient to pose significant challenges for low-resource embedded systems. We acknowledge that this approach may leave some value untapped by not also targeting larger-scale problems. However, any query-intensive application involving tens of thousands of queries or more will quickly exceed the time and memory constraints of typical devices. GPBO without MP has no inherent cap on memory or computational time, meaning that for any sufficiently powerful device, there will always be a number of queries Q that becomes infeasible.
> In practice, our method performs even better with larger query sets, and, in hindsight, we should have placed greater emphasis on this observation.
>
> 2. Lack of baselines. The authors propose reducing the data size by dropping training data randomly. However, there is no other baselines to access how good or how bad this idea is.
>
> We propose that no other GP-based optimization method can work on this problem within a fixed guarantee of execution time. However, alternative model-free methods do have fixed execution time and could have been tested here, and in a future version of this paper we could include those.
>
> 3. Question: In Line 15 of Algorithm 1, how enumerate all x in X? Is the domain discretized?
>
> Yes, here we reported having chosen to evaluate the UCB on a discrete set of points, by consequence, we only have discrete domain options.

---

### Official Review · Reviewer_1Dih · 2024-11-04

**Soundness:** 2
**Presentation:** 2
**Contribution:** 1
**Rating:** 3
**Confidence:** 4

**Summary:**

The work proposes a memory and computational cost efficient version of Bayesian Optimization algorithm by pruning the sampled data points (training data points). They work suggests various methods for pruning the data points - i) randomized pruning, ii) First in first out pruning. The work also suggests the theoretically inspired pruning method which focuses on minimal reduction of KL divergence between the updated posterior containing the point and not containing the point but does not pursue this dues to the computational costs linked with the method. Further, the computation of GP uncertainty is updated to follow the minimum of the posterior with and with out the updated point. The algorithm is tested against the benchmark algorithms and on the real world neurostimulation dataset.

**Strengths:**

The paper presents a memory and computational cost efficient BO algorithm which is tested on benchmark and real world data set.

**Weaknesses:**

Though the work presents a method for memory pruning, there is no theoretical backing for the algorithm. Also, absence of the analysis makes the algorithm less appealing.
Further, when computing the minimum of standard deviation $\min(\tilde{\sigma}(x), \sigma(x))$ the old standard deviation needs to be stored for every query which would be an additional memory cost if the old kernel matrix is stored else would be additional computational cost at each iteration if only the data points are stored.
Additionally, why is the suggested methods not computing the max of means i.e, $\max(\tilde{\mu}(x), \mu(x))$, wouldn't this result in better optimization strategy?
They flow of paper can be organized better to give more details about the algorithm. There is just one line about the randomized pruning strategy used in the algorithm, instead since this is the most novel part of the algorithm this needs to be a different subsection with the details of all the strategies tried and with insight on why the randomized strategy performed well.

**Questions:**

please look at In the weakness section.

---

> ### Author Response · Authors · 2024-11-19
>
> We would like to thank you all for spending time reading and providing feedback on our paper.
> In the following, we wish to extend our responses to the points you raised. No matter where this will lead the paper, we sought not to leave your comments and suggestions unanswered.
>
> 1. There is no theoretical backing for the algorithm. Also, absence of the analysis makes the algorithm less appealing.
>
> (Assuming absence of analysis refers to theoretical analyses, like convergence etc.). We agree with you. We cannot propose any proof e.g.of convergence for MP - even more we could prove that convergence cannot be guaranteed. The cap in the maximum execution time is a tall order for an optimization algorithm, here we propose a performant method that only comes with execution time but no convergence guarantee.
>
> 2. when computing the minimum of standard deviation the old standard deviation needs to be stored for every query which would be an additional memory cost if the old kernel matrix is stored else would be additional computational cost at each iteration if only the data points are stored.
>
> The extra memory cost is true, yet minor with respect to the cost of computing the covariance matrix during GP update, which is quadratic in memory. Storing the standard deviation is linear to the total number of points in the input space being evaluated by UCB, and equal at any time than 50% of the memory required to evaluate UCB.
>
> 3. why is the suggested methods not computing the max of means
>
> We found this suggestion very interesting and tested it on our benchmarks, however the results are unfortunately inferior. We think it may be due to - when having overestimated some local values of f - being unable to reposition these values to the proper mean with new observations; the wrong estimates simply persist indefinitely.
>
> 4. The flow of paper can be organized better to give more details about the algorithm.
>
> Thank you for your suggestion, we will improve this.

---

### Note · Authors · 2024-11-26

I have read and agree with the venue's withdrawal policy on behalf of myself and my co-authors.